# Enhanced Algorithm for the Detection of Preimpact Fall for Wearable Airbags

**DOI:** 10.3390/s20051277

**Published:** 2020-02-26

**Authors:** Haneul Jung, Bummo Koo, Jongman Kim, Taehee Kim, Yejin Nam, Youngho Kim

**Affiliations:** Department of Biomedical Engineering and Institute of Medical Engineering, Yonsei University, Seoul 03722, Korea; hanul1219@ybrl.yonsei.ac.kr (H.J.); bmk726@ybrl.yonsei.ac.kr (B.K.); jmkim0127@ybrl.yonsei.ac.kr (J.K.); xogml0923@naver.com (T.K.); namyj_1007@naver.com (Y.N.)

**Keywords:** falls, airbag, complementary filter, preimpact, threshold-based, IMU

## Abstract

Fall-related injury is a common cause of mortality among the elderly. Hip fractures are especially dangerous and can even be fatal. In this study, a threshold-based preimpact fall detection algorithm was developed for wearable airbags that minimize the impact of falls on the user’s body. Acceleration sum vector magnitude (SVM), angular velocity SVM, and vertical angle, calculated using inertial data captured from an inertial measurement unit were used to develop the algorithm. To calculate the vertical angle accurately, a complementary filter with a proportional integral controller was used to minimize integration errors and the effect of external impacts. In total, 30 healthy young men were recruited to simulate 6 types of falls and 14 activities of daily life. The developed algorithm achieved 100% sensitivity, 97.54% specificity, 98.33% accuracy, and an average lead time (i.e., the time between the fall detection and the collision) of 280.25 ± 10.29 ms with our experimental data, whereas it achieved 96.1% sensitivity, 90.5% specificity, and 92.4% accuracy with the SisFall public dataset. This paper demonstrates that the algorithm achieved a high accuracy using our experimental data, which included some highly dynamic motions that had not been tested previously.

## 1. Introduction

With the world’s population ageing, there is an increasing focus on age-related health issues. Fall-related injuries are the primary cause of injury and mortality in the elderly. According to the WHO, every year, 37.3 million people are injured severely enough to receive medical treatment, and 646,000 people die due to falls. Adults over 65 years of age suffer the greatest number of fatal falls [1]. Reduced levels of bone density, decreased muscle mass, as well as slow reflexes increase the risk of fall-related injuries in the elderly [2,3]. Hip fractures are considered the most dangerous among various injuries, as they can reduce mobility and lead to many complications, and even death [4]. In addition, there is a social dimension to this issue, because it mandates special care for the elderly. From an economic perspective, fall-related medical expenses in the elderly were reported to have accounted for as much as USD 50 billion for nonfatal falls and 754 million for fatal falls in the year 2015 alone [5]. Therefore, falls and fall-related injuries are major sources of physical, social, and economic problems among the elderly.

In the past, many studies have tried to improve the physical performance of the elderly by implementing exercise programs to help prevent falls. Lord et al. [6] conducted a fall prevention exercise program based on physiological profile assessments and sit-to-stand tests. They found that while the program improved knee flexion and visual ability, there was no significant reduction in fall risk. Røyset et al. [7] also conducted a fall prevention program using the Norwegian version of the fall risk assessment method, “STRATIRY” (score 0–5), but achieved no significant improvement statistically when compared to the control group.

There are many commercially available hip protection pads that help mitigate the impact of a fall, but they are inconvenient to use in daily life as they must be worn under the clothes. Additionally, their performance varies, as they tend to slide away from the correct position when worn over time [8,9]. To ensure better impact attenuation during falls, some researchers have also developed wearable airbags [10,11,12]. Jeong et al. [13] developed a hip impact simulator and compared the impact attenuation performance of hip protection pads with a wearable airbag. Their results showed that the wearable airbag did a better job of cushioning the impact from a fall because of its deformation at the time of impact, which increased the contact time and area involved during the impact. However, to successfully utilize a wearable airbag, a fall detection algorithm with a high level of accuracy and sufficient lead time to inflate the airbag before the impact must be used.

Falls can be detected either through stationary devices, such as cameras, or wearable sensors. Rougier et al. [14] used a particle filter to track human head position in three dimensions with a single camera, and detected falls using the vertical displacement and velocity of the head. Bian et al. [15] tracked the head and waist using a depth camera to detect falls. They applied a support vector machine classifier and achieved 95.8% sensitivity, 100% specificity, and 97.9% accuracy. Wang et al. [16] created a surveillance video–based fall detection system using PCANet and a linear support vector machine classifier that achieved 88.87% sensitivity and 98.9% specificity. Li et al. [17] used Kinect sensors to distinguish falls based on whether the zero moment point was within the dynamic supporting area or not; they obtained 100% sensitivity, 81.3% specificity, and 91.7% accuracy. However, detecting falls using such stationary camera devices has some drawbacks, including limited coverage area and privacy issues.

In contrast, many fall detection systems have been developed that use wearable sensors, such as inertial measurement unit (IMU) and electromyography (EMG) sensors, based on machine learning (ML) and threshold-based approaches. Rescio et al. [18] attached an EMG sensor to the soleus muscle to detect falls and selected the optimal feature among ten types of time-domain features using a Markov random field-based Fisher-Markov selector. They achieved 89.5% sensitivity and 91.9% specificity. Yoo et al. [19] obtained acceleration properties from an IMU positioned on the wrist that detects falls using an artificial neural network, and achieved 100% accuracy. Aziz et al. [20] used 3-axis acceleration data from an IMU attached on the waist, and compared its performance between logistic regression, decision tree, Naïve Bayes, K-nearest neighbor, and support vector machine. Among them, the support vector machine showed the best performance, with 96% sensitivity and 96% specificity. In general, ML-based fall detection algorithms show good performance, but it takes considerable time to extract features and train classifiers. Therefore, ML-based algorithms are not appropriate for preimpact fall detection where a rapid detection time is required. Therefore, threshold-based fall detection algorithms would be more suitable for predicting falls because of their low computational requirements and quick response time.

Thanh et al. [10] captured 3-axis acceleration data from the waist position, to which they applied a Kalman filter to reduce measurement and sensor errors. Their threshold-based fall detection algorithm used the root mean square of acceleration, vertical angle, and vertical velocity, and achieved 100% fall detection accuracy. Zhong et al. [11] proposed a threshold-based fall detection algorithm using vertical displacement and vertical velocity from an IMU sensor on the front waist. They obtained 93.6% sensitivity and 95.6% specificity with an average lead time of 363 ms. Nyan et al. [21] developed a preimpact fall detection algorithm that used angles calculated from the torso and thigh (with the help of 2 IMUs), as well as the correlation coefficients of the torso and thigh angles. The algorithm achieved 100% specificity, 95.2% specificity, and a lead time of 700 ms. Wang et al. [22] calculated acceleration magnitude, acceleration cubic-product-root magnitude, and angular velocity cubic-product-root magnitude from the acceleration and angular velocity data obtained from an IMU placed on the front chest to develop a threshold-based fall detection algorithm. In the study, the algorithm was evaluated using their experimental data and two different public datasets (Cogent Labs and UMAFall). They achieved 98.9%, 98.0%, and 96.6% sensitivity and 100%, 96.6%, and 83.2% specificity based on their experimental data, cogent labs dataset, and UMAFall dataset, respectively. Ahn et al. [23] developed a threshold-based fall detection algorithm based on acceleration SVM, angular velocity SVM, and triangle feature calculated from acceleration and angular velocity data from a sensor placed on the front waist. They reported 100% sensitivity and 83.9% specificity using the SisFall public dataset. While some of the previously discussed threshold-based fall detection algorithms showed high detection accuracies, they would not be practical in real-life applications because only less-dynamic (walking, standing, sitting, etc.) or typical motions (climbing upstairs/downstairs, etc.) were included in these studies.

In this study, 6 types of simulated falls and 14 activities of daily life (ADLs), including some highly dynamic motions, were selected to simulate real-life situations. To capture inertial data, an IMU was used. From the captured inertial data, acceleration SVM, angular velocity SVM, and vertical angle were calculated using complementary filter with a proportional integral (PI) controller. Subsequently, a threshold-based preimpact fall detection algorithm was developed based on these parameters. The algorithm’s performance was evaluated in a lab experiment involving 30 research participants. For an objective evaluation, we evaluated the algorithm using the SisFall public dataset as well [24].

## 2. Materials and Methods

### 2.1. Participants

Thirty healthy young men (age: 23.4 ± 1.2 years, height: 173.7 ± 4.75 cm, and weight: 74.8 ± 8.41 kg) were recruited to perform ADLs and simulated falls. None of the participants had any neuromusculoskeletal abnormalities. The experiment was approved by the Yonsei University Research Ethics Committee (1041849-201811-BM-112-01) and conducted with written consent from all participants.

### 2.2. Equipment

The MPU-9250 [25] (InvenSense, San Jose, USA) sensor, measuring 3-axis acceleration (±16 g), 3-axis angular velocity (±2000 °/s), and 3-axis magnetism (±4800 μT), has been generally used for motion capturing and gesture recognition [26,27]. The nRF52832 (Nordic Semiconductor, Trondheim, Norway) was used to receive the inertial data from MPU-9250 through I2C communication and to send the data to PC via radio frequency (RF; 2.4 GHz) communication. LabView (National Instruments, Austin, USA) was used to save the data on the PC. The entire workflow is shown in Figure 1. Synchronized with the IMU sensor which was sampled at 100 Hz, a webcam and a Bonita high-speed camera (Vicon Motion System Ltd., Oxford, UK) were used to determine the time when the body collided with the ground in fall motions.

The calibration of the IMU sensor was done at angular velocities of 30, 60, 180, 360, and 600°/s in roll, pitch and yaw axis directions using a gimbal device before performing the experiment (Figure 2). As a result, it was confirmed that the angular velocities measured by the sensor had errors of 2.74 ± 1.91%, 1.30 ± 0.79%, 0.49% ± 0.36%, 0.50 ± 0.35%, and 0.48 ± 0.22% at the given values of 30, 60, 180, 360, and 600°/s, respectively. In addition, the acceleration measured by the sensor matched very well with gravity.

### 2.3. Experimental Procedures

Using a rubber band, an IMU was positioned in the middle of the left posterior superior iliac spine (LPSIS) and the right posterior superior iliac spine (RPSIS), which was close to the center of mass of human and not affected according to the personal obesity (Figure 3). The axes of the IMU were set so that the anteroposterior was the x-axis, the mediolateral the y-axis, and the superoinferior the z-axis. All participants performed 14 motions simulating ADLs and 6 motions simulating falls, 3 times each (Table 1). Pictures of some ADLs and fall motions experiments are shown in Figure 4. After performing each motion, the sensor location was checked for the correct position. As a result, it was confirmed that the position of the sensor did not change much when performing each motion. Among ADLs, movements in which peak acceleration SVM was less than 2 g were defined as less-dynamic, while others were defined as highly dynamic. ADLs, including less-dynamic motions (walking, squatting, waist bending, etc.) and highly dynamic motions (quickly sitting on the chair and getting up, trying to get up but collapsing into the chair, etc.) were selected based on Ahn et al. [12], Sucerquia et al. [24], and Eduardo et al. [28]. Fall simulations included backward and sideways falls, but not forward-facing falls, as the hip does not directly collide with the floor in such falls. Fall simulations were performed on a 40 cm thick mattress for the safety of the participants.

### 2.4. Data Analysis

The 3-axis acceleration and angular velocity data obtained from the IMU was saved as a csv file on a PC using LabView, and data analysis was performed using MATLAB R2019a (MathWorks Inc., Massachusetts, USA). A 5 Hz digital low-pass filter was applied to the inertial data to remove high frequency noise. Then, acceleration SVM $(a_{SVM}$) and the angular velocity SVM ($\omega_{SVM}$) were calculated as
(1)$$a_{SVM}=\sqrt{a_{X}^{2}+a_{Y}^{2}+a_{Z}^{2}},$$
(2)$$\;\omega_{SVM}=\sqrt{\omega_{X}^{2}+\omega_{Y}^{2}+\omega_{Z}^{2}},\;$$
where $a_{X},\;a_{Y}$, and $a_{Z}$ represent accelerations in the *x*-, *y*-, and *z*-axis, respectively, and $\omega_{Z},\;\omega_{Y}$, and $\omega_{Z}$ are the angular velocities in the *x*-, *y*-, and *z*-axis, respectively.

In this study, a complementary filter was used to obtain an accurate vertical angle in both static and dynamic conditions by compensating the vertical angle calculated using 3-axis angular velocity with the vertical angle calculated using 3-axis acceleration. In addition, the PI controller was used to minimize integration errors and the effect of external impacts. A flow chart showing the vertical angle calculation using the complementary filter with a PI controller is shown in Figure 5 [29].

The Euler angular velocities of roll and pitch, $\dot{\emptyset_{G}\;}$ and $\dot{\theta_{G}}$, were calculated as
(3)$$\left[\begin{array}{c} \dot{\emptyset_{G}}\\ \dot{\theta_{G}} \end{array}\right]=\left[\begin{array}{ccc} 1 & sin\emptyset tan\theta & cos\emptyset tan\theta \\ 0 & cos\emptyset & -sin\emptyset \end{array}\right]\left[\begin{array}{c} \omega_{X}\\ \omega_{Y}\\ \omega_{Z} \end{array}\right],$$
where $\omega_{X}$, $\omega_{Y}$, and $\omega_{Z}$ are *x*-, *y*-, and *z*-axis angular velocities, respectively, measured from the IMU, and ∅ and θ  are the previous vertical angles of roll and pitch, respectively.

The vertical angles of roll and pitch, $\emptyset_{A}$ and $\theta_{A}$, from 3-axis acceleration data were calculated as
(4)$$\emptyset_{A}=\;\tan^{-1}\frac{a_{Y}}{\sqrt{a_{X}^{2}+a_{Z}^{2}}},$$
(5)$$\;\theta_{A}=\tan^{-1}\frac{a_{X}}{\sqrt{a_{Y}^{2}+a_{Z}^{2}}},\;$$
where $a_{X},\;a_{Y}$, and $a_{Z}$ represent accelerations in the *x*-, *y*-, and *z*-axes, respectively.

In order to validate the calculation of vertical angles, a trigonal prism was built with angles of 30°, 60°, and 90°. The IMU was placed on its surfaces and vertical angles were calculated using the complementary filter with PI controller. The results showed that the calculated vertical angles were in good agreement with the true values with errors of 1.57 ± 1.59°.

### 2.5. Preimpact Fall Detection Algorithm

Based on the performed experiment, four threshold values (acceleration SVM, angular velocity SVM, roll angle, and pitch angle) were selected to develop a threshold-based preimpact fall detection algorithm. Sensitivity, specificity, and accuracy were calculated to evaluate the performance of the algorithm. In this study, the threshold values were set to maximize accuracy with 100% sensitivity. Sensitivity, specificity, and accuracy were calculated as
(6)$$\mathrm{Sensitivity}\;\left(\%\right)=\frac{True\;positives}{True\;positives\;+\;False\;negatives\;}\times 100,$$
(7)$$\;\mathrm{Specificity}\;\left(\%\right)=\frac{True\;negatives}{True\;negatives\;+\;False\;positives}\times 100,\;$$
(8)$$\;\mathrm{Accuracy}\;\left(\%\right)=\frac{True\;positives\;+\;True\;negatives}{True\;positives\;+\;True\;negatives\;+\;False\;positives\;+\;False\;negatives}\times 100,$$
where True positives is the number of falls detected as falls, False positives is the number of ADLs detected as falls, True negatives is the number of ADLs detected as ADLs, and False negatives is the number of falls detected as ADLs.

In general, highly dynamic motions have large acceleration and angular velocity. However, vertical angles in ADLs are relatively different from those in fall motions. In the present algorithm, two vertical angles (roll and pitch) were used. Roll and pitch angles were defined as the vertical angles of the body in the frontal and sagittal plane, respectively. Because the roll angle does not change significantly in ADLs, its threshold was set to 28°. The threshold of pitch angle was set to 45°, since the upper body moved back and forth in some ADLs. As mentioned earlier, forward falls were not considered in this study, and thus, the absolute value of the pitch angle was not used in the algorithm. Figure 6 and Figure 7 represent the roll and pitch angles of less-dynamic, highly dynamic, and fall. In lying motions, the vertical angle changed to about 90° as in fall motions, but acceleration SVM did not. Using the grid search method to maximize the specificity with 100% sensitivity, the threshold of acceleration SVM was set to 0.82 g. Figure 8 represents acceleration SVMs of less-dynamic, highly dynamic, and fall motions. The threshold of angular velocity SVM was set to 47.3 °/s [23]. Figure 9 represents angular velocity SVMs of less-dynamic, highly dynamic, and fall motions. In Figure 6, Figure 7, Figure 8 and Figure 9, threshold values are shown as horizontal blue lines and impact time of fall motions are shown with a vertical red line. Figure 10 represents a flow chart of the algorithm.

## 3. Results

The algorithm developed in this study was evaluated using experimental data from the 30 participants. In so doing, we obtained 100% sensitivity, 97.54% specificity, and 98.33% accuracy. However, a few ADLs were detected as falls, especially highly dynamic motions such as jogging and quickly sitting on a low-height mattress and getting up. The numbers of false positives are depicted in Table 2.

As it is important to detect falls before the actual impact, lead time (the interval between fall detection time and collision time) was calculated as shown below:Lead time = Collision time − Fall detection time(9)

The average lead time was 280.25 ± 10.29 ms. The lead times of all six types of fall motions are shown in Table 3. In case of a sit down-backward fall, the lead time was shorter than with other falls because the height of the fall was relatively low. The inflation time of the wearable airbag was approximately 200 ms, since a spring-triggered type inflator instead of a gunpowder type one was used [23]. Therefore, the airbag could be inflated sufficiently before impact using the developed algorithm.

In addition, the SisFall public dataset was used to objectively evaluate the proposed algorithm, and its performance was compared with other similar studies [23,30,31,32]. As can be seen in Table 4, our proposed algorithm achieved the highest accuracy (92.4%) (with 96.1% sensitivity and 90.5% specificity). In case of the SisFall dataset, the lead time could not be calculated because there were no video data available.

## 4. Discussion

Some previous studies reported very high accuracies for their threshold-based fall detection algorithms. As discussed previously, Nyan et al. [21] achieved 100% specificity, 95.2% sensitivity, and an average lead time of 700 ms. However, their study included only less-dynamic motions (such as walking, lying down, and ascending/descending stairs) as ADLs. If the algorithm had been evaluated against highly dynamic motions, the specificity achieved might have been significantly lower. Similarly, Ahn et al. [12] conducted simulated ADLs and falls with 40 healthy young men, and developed a threshold-based preimpact fall detection algorithm using acceleration SVM, angular velocity SVM, and vertical angle. Although they claimed 100% accuracy and a 401.9 ± 46.9 ms lead time, the ADLs performed in the study did not include highly dynamic motions such as quickly sitting on the chair and getting up, or quickly sitting on a low-height mattress and getting up. When evaluating the algorithm with our experimental data, which included highly dynamic motions, 100% sensitivity and 58.17% specificity were achieved. The reason for the significant decrease in the specificity is that vertical angle as well as acceleration are heavily affected in highly dynamic motions. To counter this, a complementary filter with a PI controller was used to calculate accurate vertical angle values which resulted in 100% sensitivity and 97.54% specificity with our experimental data, which included highly dynamic motions.

Despite achieving 100% sensitivity with our experimental data, there were 2.46% false positives (detecting ADLs as fall): 9 (jogging), 1 (running in place), 1 (lying down), 6 (quickly sitting on the chair and getting up), 4 (trying to get up and collapsing into the chair), and 7 (quickly sitting on low-height mattress and getting up). The jogging track used during the experiment involved curve sections where the body tilted laterally, and thus, the roll angle exceeded the threshold. Similarly, during running in place, the repetitive impact changed the axis of IMU and the vertical angle exceeded the threshold. While lying down, the vertical angle always exceeded the threshold. When suddenly moving in this condition, the acceleration SVM and angular velocity SVM exceeded the thresholds. In other such false positives, the impact was similar to one of a fall (ex., because of collapsing into a chair or mattress), where the pitch angle exceeded the threshold when the upper body moved a little further back. Even in the SisFall public dataset, running (23/46) and lying (102/230) motions were often incorrectly detected as falls. However, it needs to be noted that our results are based on experimental data from young participants. The movement of the elderly is slower; therefore, the actual specificity of our algorithm would improve when detecting their slower jogging motions. However, the motion of lying is not very different between the young and the elderly; here, other parameters like vertical velocity need to be used to reduce detection errors.

Our algorithm achieved an accuracy of 92.4% using the SisFall public dataset and showed the best performance when compared with other similar algorithms. However, in any such fall detection algorithm, 100% sensitivity is essential; otherwise, it could lead to serious injuries to the user if the airbag does not inflate when a fall occurs. We found that our algorithm detected some falls as ADLs in cases of a motion that simulates falling while sitting. A similar motion was also present in our experiment, where our algorithm showed a 100% detection rate. The reason for this difference could be that our algorithm was developed based on the IMU being located on the middle of the LPSIS and RPSIS, whereas the SisFall dataset was prepared with the IMU placed in the middle of the LASIS and RASIS.

In future study, additional features such as vertical velocity and the acceleration and angular velocity of each axis will be considered to reduce the false detection rate. In addition, since only young participants were recruited in this study, future experiment will be performed with older participants, whereby ages more closely aligned with the target user groups would be obtained.

## 5. Conclusions

In this study, a threshold-based preimpact fall detection algorithm using an IMU placed in the middle of the LPSIS and RPSIS was developed. The vertical angle was considered the most important value to detect falls; therefore, it was calculated using a complementary filter with a PI controller to minimize integration errors and the effect of external impacts. With our experimental data, which included highly dynamic motions, 100% sensitivity, 97.54% specificity, and 98.33% accuracy were achieved. The average lead time for fall detection was 280.25 ± 10.29 ms. The algorithm was also objectively evaluated with the SisFall public dataset, with which we achieved 96.1% sensitivity, 90.5% specificity, and 92.4% accuracy. The accuracy achieved was the highest when compared with four other previous studies. Although the present algorithm is not perfect for use with a wearable airbag in real-life applications, it will be improved further in the future.

## Figures and Tables

**Figure 1 sensors-20-01277-f001:**
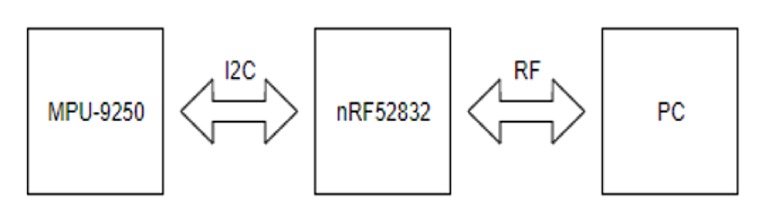
Data collection workflow.

**Figure 2 sensors-20-01277-f002:**
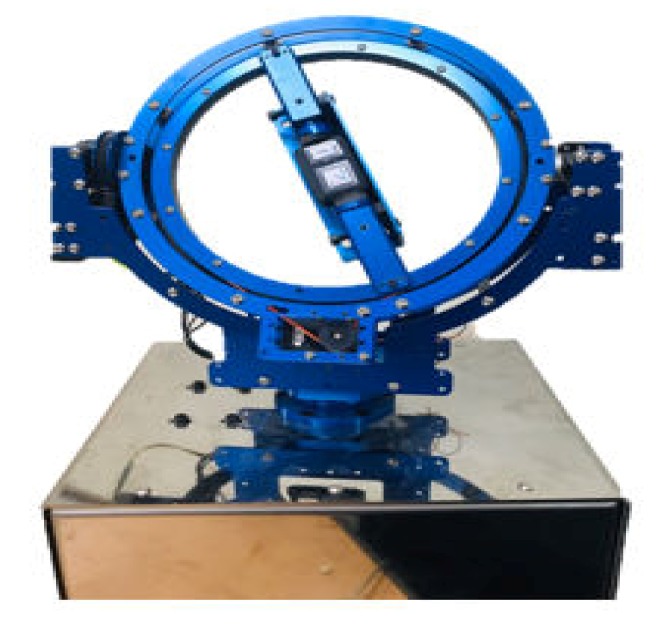
Gimbal device.

**Figure 3 sensors-20-01277-f003:**
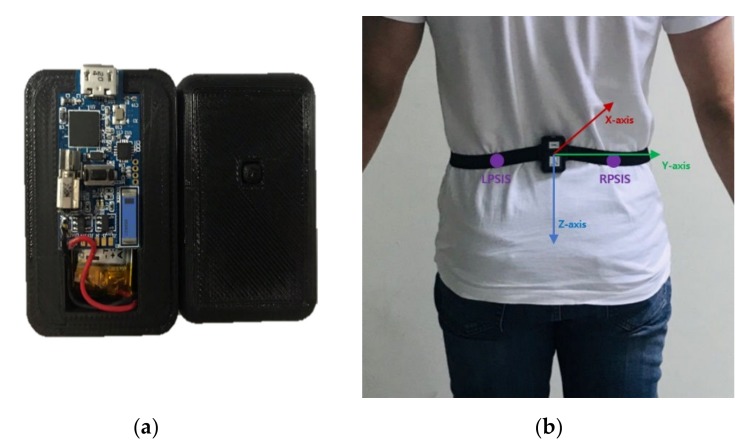
(**a**) IMU and (**b**) sensor position.

**Figure 4 sensors-20-01277-f004:**
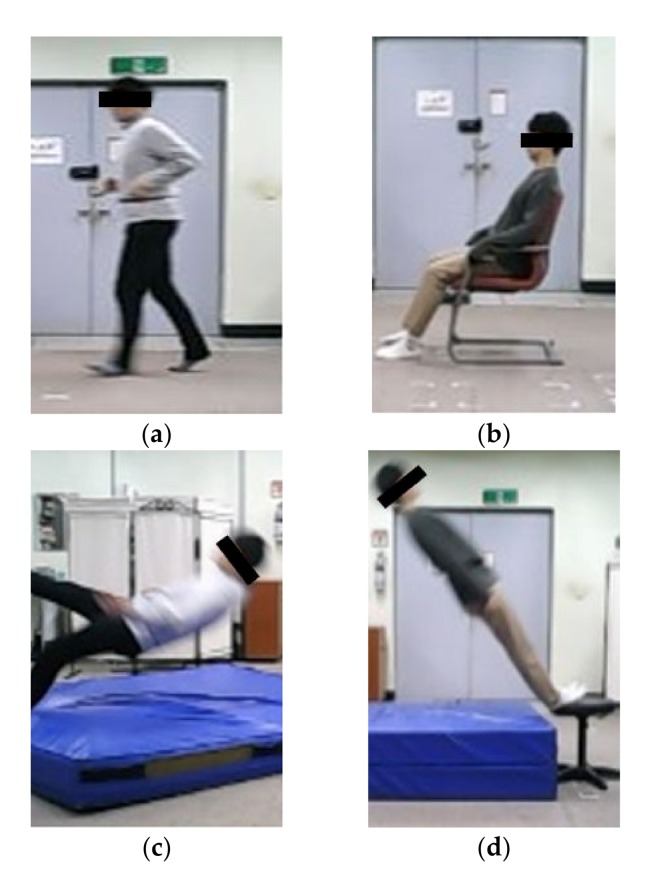
ADLs and fall motions: (**a**) Jogging, (**b**) Quickly sitting on the chair and getting up, (**c**) Slip-backward fall and (**d**) Backward fall.

**Figure 5 sensors-20-01277-f005:**
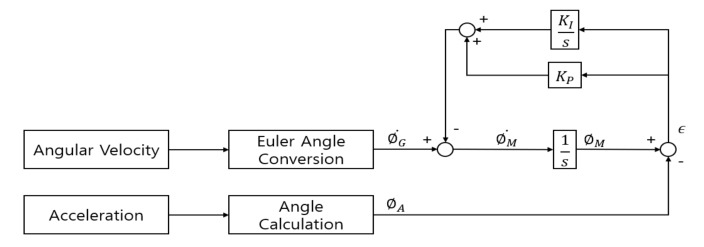
Gyroscope-Accelerometer fusion complementary filter with a PI controller.

**Figure 6 sensors-20-01277-f006:**
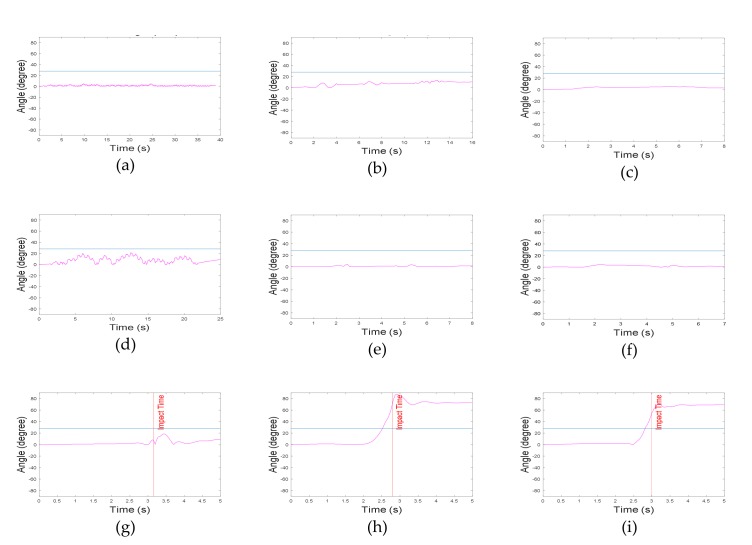
Roll angle graphs. Less-dynamic motions: (**a**) walking, (**b**) climbing up and down the stairs, and (**c**) slowly sitting the on stool and getting up. Highly dynamic motions: (**d**) jogging, (**e**) quickly sitting on the chair and getting up, and (**f**) quickly sitting on low-height mattress and getting up. Fall motions: (**g**) backward fall, (**h**) lateral fall, and (**i**) twist fall.

**Figure 7 sensors-20-01277-f007:**
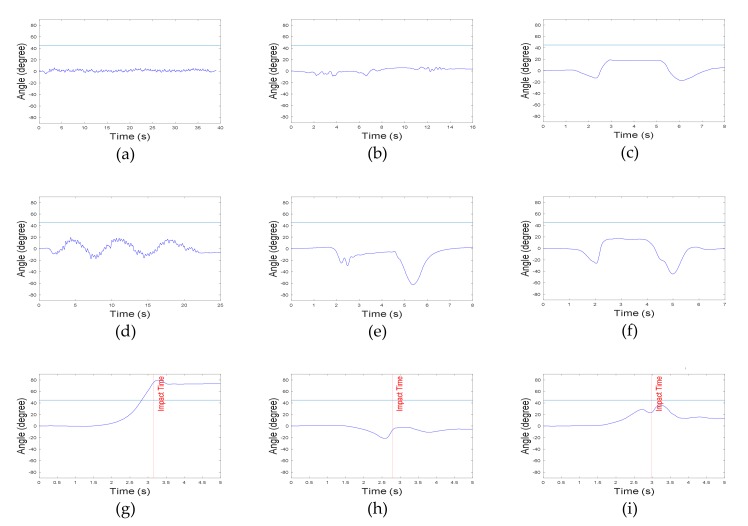
Pitch angle graphs. Less-dynamic motions: (**a**) walking, (**b**) climbing up and down the stairs, and (**c**) slowly sitting on the stool and getting up. Highly dynamic motions: (**d**) jogging, (**e**) quickly sitting on the chair and getting up, and (**f**) quickly sitting on low-height mattress and getting up. Fall motions: (**g**) backward fall, (**h**) lateral fall, and (**i**) twist fall.

**Figure 8 sensors-20-01277-f008:**
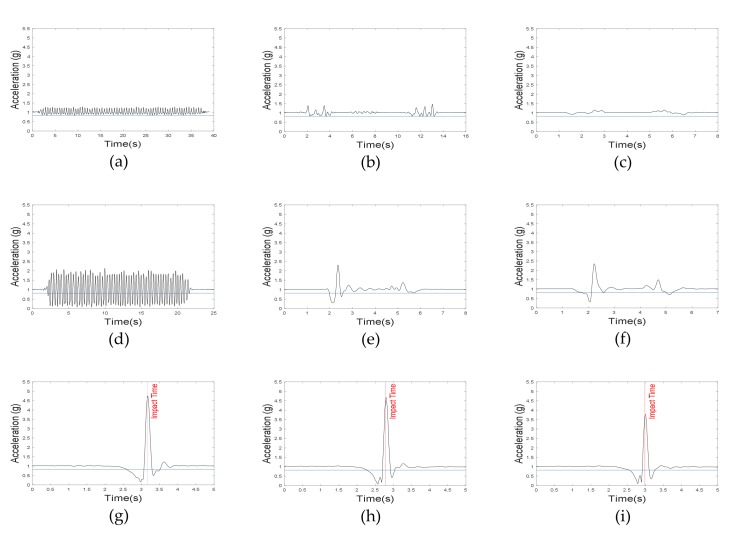
Acceleration SVM graphs. Less-dynamic motions: (**a**) walking, (**b**) climbing up and down the stairs, and (**c**) slowly sitting on the stool and getting up. Highly dynamic motions: (**d**) jogging, (**e**) quickly sitting on the chair and getting up, and (**f**) quickly sitting on low-height mattress and getting up. Fall motions: (**g**) backward fall, (**h**) lateral fall, and (**i**) twist fall.

**Figure 9 sensors-20-01277-f009:**
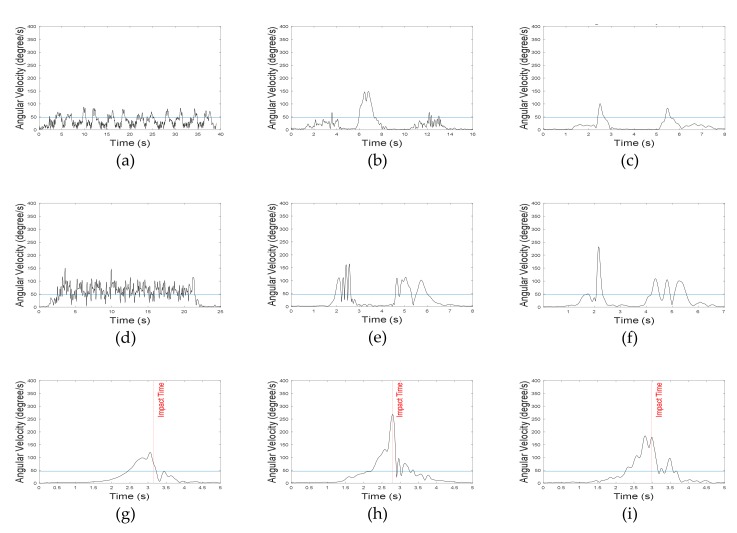
Angular velocity SVM graphs. Less-dynamic motions: (**a**) walking, (**b**) climbing up and down the stairs, and (**c**) slowly sitting on the stool and getting up. Highly dynamic motions: (**d**) jogging, (**e**) quickly sitting on the chair and getting up, and (**f**) quickly sitting on low-height mattress and getting up. Fall motions: (**g**) backward fall, (**h**) lateral fall, and (**i**) twist fall.

**Figure 10 sensors-20-01277-f010:**
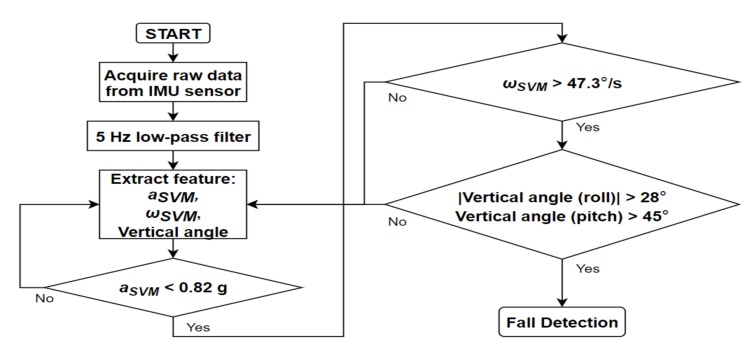
Flow chart of the proposed algorithm.

**Table 1 sensors-20-01277-t001:** List of activities.

Activities		
**ADLs**	1. Walking	8. Climbing up and down the stairs
	2. Jogging *	9. Slowly sitting on the stool and getting up
	3. Squatting	10. Quickly sitting on the chair and getting up *
	4. Waist bending	11. Trying to get up but collapsing into the chair *
	5. Lying down	12. Stumbling while walking *
	6. Running in place *	13. Slowly sitting on low-height mattress and getting up
	7. Jumping *	14. Quickly sitting on low-height mattress and getting up *
**Falls**	1. Slip-backward fall	4. Backward fall
	2. Sit down-backward fall	5. Lateral fall
	3. Sit-backward fall	6. Twist fall (Back → Side)

* Highly dynamic motions (peak acceleration SVM > 2 g).

**Table 2 sensors-20-01277-t002:** Number of false positives that occurred during ADLs.

ADLs	Number of FPs	ADLs	Number of FPs
Walking	0/30	Stumbling while walking	0/90
Jogging	9/30	Climbing up and down the stairs	0/90
Squatting	0/90	Slowly sitting on the stool and getting up	0/90
Waist bending	0/90	Quickly sitting on the chair and getting up	6/90
Running in place	1/90	Trying to get up but collapsing into the chair	4/90
Jumping	0/90	Slowly sitting on low-height mattress and getting up	0/90
Lying	1/90	Quickly sitting on low-height mattress and getting up	7/90

**Table 3 sensors-20-01277-t003:** Lead times according to type of falls.

Type of Falls	Lead Time (ms)	Type of Falls	Lead Time (ms)
Slip-backward fall	292 ± 10.61	Backward fall	333 ± 9.71
Sit down-backward fall	151 ± 5.37	Lateral fall	335 ± 8.26
Sit-backward fall	296 ± 5.38	Twist fall (Back → Side)	274 ± 8.28

**Table 4 sensors-20-01277-t004:** Performance comparison with other similar studies.

	Bourke et al. [30]	Wu et al. [31]	Tamura et al. [32]	Ahn et al. [23]	This Study
**Accuracy (%)**	87.2	80.5	81.8	90.3	92.4
**Sensitivity (%)**	100	100	93	100	96.1
**Specificity (%)**	78.7	67.6	74.4	83.9	90.5

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
