# Peer review of "Enhanced Algorithm for the Detection of Preimpact Fall for Wearable Airbags"

_sensors, 2020, doi:10.3390/s20051277_

Round 1

Reviewer 1 Report

Sugestion for the title:

Enhanced algorithm for detection of the pre-impact fall threshold used on wearable airbags

Most of the references contains articles on human fall detection. The bibliography does not contain references regarding the characteristics of the sensors, their accuracy, repeatability... I have not found details about the position of the equipment on the human body. Is this position controlled? Have you made any determinations regarding the positioning of the sensor on the same human body in several cases? Is its position within a controlled range? What is the probability that the sensor will be positioned in the same place? What influence on the results does the positioning accuracy of the sensor on the human body? What can you say about repeatability? Have you built a simple equipment, to control the angles and accelerations, on which to mount the sensor to see how it works? Not just mounted on human subjects. I think the sensor need to be calibrated before use. In the paper, I did not find details about this operation. During the researches, did a study on precision of measurements and repeatability? Should the use of other sensors be feasible? Specify what data you get from the sensors, the board, the corresponding units of measurement and the coordinate systems. What is the probability that the system will malfunction when the patient is in a lift, bus, train (in a system that has high accelerations)? Could you put more pictures during the experiments?

Author Response

Point 1: Suggestion for the title: Enhanced algorithm for detection of the pre-impact fall threshold used on wearable airbags

Response 1: Thank you for your suggestion. We modified the title, based on your suggestion.

(page 1)

Enhanced Algorithm for the Detection of Pre-impact Fall used for Wearable Airbags

Point 2: Most of the references contains articles on human fall detection. The bibliography does not contain references regarding the characteristics of the sensors, their accuracy, repeatability.

Response 2: We added three references (#25-#27) regarding the characteristic of MPU-9250 sensor and its application. You can find them on 118-120 lines (page 3) and 373-378 lines (page 13). The reference numbers were also rearranged.

(page 3)

...

2.2. Equipment

*MPU-9250 [25] (InvenSense, USA) sensor, measuring 3-axis acceleration (±16 g), 3-axis angular velocity (±2000 °/s), and 3-axis magnetism (±4800 μT), has been generally used for motion capturing and gesture recognition [26,27].*

...

(page 13)

References

...

25. InvenSense, MPU-9250 Nine-Axis MEMS Motion Tracking Device, 2015, https://www.invensense.com/products/motion-tracking/9-axis/mpu-9250/

26. Kromemwett, N.; Ruppelt, J.; Trommer, G. F. Motion monitoring based on a finite state machine for precise indoor localization. Gyroscopy and Navigation 2017, 8(3), pp. 190-199

27. Fang, B.; Sun, F.; Liu, H.; Liu, C. 3D human gesture capturing and recognition by the IMMU-based data glove. Neurocomputing 2018, 277, pp. 198-207

...

Point 3: I have not found details about the position of the equipment on the human body. Is this position controlled? Have you made any determinations regarding the positioning of the sensor on the same human body in several cases? Is its position within a controlled range? What is the probability that the sensor will be positioned in the same place? What influence on the results does the positioning accuracy of the sensor on the human body? What can you say about repeatability?

Response 3: We mentioned the position of the sensor in page 4 (137-139 lines).

(page 3)

2.3. Experimental Procedures

*Using the rubber band, an IMU was positioned in the middle of the left posterior superior iliac spine (LPSIS) and the right posterior superior iliac spine (RPSIS), which was close to the center of mass of human and not affected according to the personal obesity (Figure 2).*

...

The sensor position was confirmed after each measurement. (143-144 lines, page 4)

(page 4)

2.3. Experimental Procedures

...

*After performing each motion, the sensor location was checked for the correct position. As a result, it was confirmed that the position of the sensor did not change much when performing each motion.*

...

Point 4: Have you built a simple equipment, to control the angles and accelerations, on which to mount the sensor to see how it works? Not just mounted on human subjects. I think the sensor need to be calibrated before use. In the paper, I did not find details about this operation. During the researches, did a study on precision of measurements and repeatability?

Response 4: We added contents related to your comments (126-131 lines in page 3, 177-180 lines in page 6). The sensor specification provided by the manufacturer stated that the sensor tolerance is 3%. You can find from reference #25.

(page 3)

2.2. Equipment

...

*The calibration of the IMU sensor was done at angular velocities of 30, 60, 180, 360 and 600°/s in roll, pitch and yaw axis directions using a gimbal device before performing the experiment (Figure 5a). As a result, it was confirmed that the angular velocities measured by the sensor had errors of 2.74±1.91%, 1.30±0.79%, 0.49%±0.36%, 0.50±0.35% and 0.48±0.22% at the given values of 30, 60, 180, 360 and 600°/s, respectively. In addition, the acceleration measured by the sensor matched very well with the gravity.*

...

(page 6)

...

Figure 5. (a) Gimbal device

...

(page 6)

2.4. Data Analysis

...

*In order to validate the calculation of vertical angles, a trigonal prism was built in which has angles of 30°, 60° and 90° (Figure 5b). The IMU was placed on its surfaces and vertical angles were calculated using the complementary filter with PI controller. Results showed that the calculated vertical angles had a good agreement with the true values with errors of 1.57±1.59°.*

Figure 5. (b) trigonal prism

...

Point 5: Should the use of other sensors be feasible? Specify what data you get from the sensors, the board, the corresponding units of measurement and the coordinate systems.

Response 5: In this study, The measuring range of MPU-9250 were ±16 g for acceleration, ±2000 °/sec for angular velocity, and ±4000 μT for magnitism. The coordinate system of the IMU sensor was mentioned in page 4 (139-141 lines).

(page 4)

2.3. Experimental Procedures

...

The axes of the IMU were set so that the anteroposterior was the x-axis, the mediolateral the y-axis, and the superoinferior the z-axis.

...

Any validated IMU sensor with the proper measuring range would be compatible for the present algorithm.

Point 6: What is the probability that the system will malfunction when the patient is in a lift, bus, train (in a system that has high accelerations)?

Response 6: As long as the vertical angle and angular velocity are within the pre-determined thresholds despite of high acceleration, no fall would be determined. We think that the algorithm developed in this study will not malfunction in lift, bus, train, etc. However, it would be necessary to evaluate the performance of the present algorithm by performing experiments in lift, bus and train (in a system that has high accelerations).

Point 7: Could you put more pictures during the experiments?

Response 7: Some pictures of ADLs and fall motions were added in the revised manuscript. You can find them on page 5.

(page 5)

2.3. Experimental Procedures

...

Figure 3. ADLs and fall motions: (a) Jogging, (b) Quickly sitting on the chair and getting up, (c) Slip-backward fall and (d) Backward fall.

Reviewer 2 Report

The paper describes a method for fall detection based on IMU. The method is mean to be used together with a wearable airbag for preventing hip injuries in elderly falls. The results show a good performance in terms of performance and computation time.

The paper is well written and structured. The method is clearly described and the results are well analyzed.

While the technical results of the paper seem to be good compared with previous works, the method itself doesn't present any big advancement with respect to the state of the art. The main concern is regarding the selection of the thresholds. It is understandable that the proposed method, being simpler than other ones, is also faster and can be easily implemented. However, the selection of the thresholds could be done in a more sophisticated way. What was the criteria used to select the thresholds? Was it just empirically? There is only one mention to the acceleration SVM ("selected by comparing the graphs of ADLs and fall motions). With all the data available (collected and openly available), there are a lot of methods that could be used to optimize the thresholds to maximize the performance of the algorithm, without affecting the computation time.

There are some minor comments:

 - In the abstract, the "lead time" is not defined. Just a short description, such as time lapse between the fall detection and the contact, would help a reader new in the field.

 - The Support Vector Machine could be defined as SVM, since it appears several times in the text (introduction). However, seeing that also Sum Vector Magnitude (SVM) is defined with the same acronym, it is reasonable to define only the second one which is more relevant for the paper. However, at the end of the introduction the acronym is defined twice.

 - In section 2.2, it is not clear how the high-speed camera is used. Vicon is known for motion tracking, so it is not clear why using the Vicon if in the paper is not described any use of a motion tracking system. In Figure 2 for example, there are no markers on the subject, so it is reasonable to think that no motion tracking is used. If the Vicon camera is used not for motion tracking but for other purposes, it would be good to specify it explicitly.

Author Response

Point 1: The paper describes a method for fall detection based on IMU. The method is mean to be used together with a wearable airbag for preventing hip injuries in elderly falls. The results show a good performance in terms of performance and computation time.

The paper is well written and structured. The method is clearly described and the results are well analyzed.

Response 1: Thank you for your kind comments.

Point 2: While the technical results of the paper seem to be good compared with previous works, the method itself doesn't present any big advancement with respect to the state of the art.

Response 2: Many studies have been conducted to classify falls and ADLs based on threshold values. However, most studies did not address highly dynamic motions in which many previous algorithms malfunctioned. In this study, some highly dynamic motions such as jogging, jumping, quickly sitting on the chair and getting up and stumbling while walking were included for the experiment, and a complementary filter with a proportional integrator controller was used to accurately calculate the vertical angle of human. The accurate vertical angle was obtained even in highly dynamic motions, which resulted in the robust fall detection with high accuracy.

Point 3: The main concern is regarding the selection of the thresholds. It is understandable that the proposed method, being simpler than other ones, is also faster and can be easily implemented. However, the selection of the thresholds could be done in a more sophisticated way. What was the criteria used to select the thresholds? Was it just empirically? There is only one mention to the acceleration SVM ("selected by comparing the graphs of ADLs and fall motions). With all the data available (collected and openly available), there are a lot of methods that could be used to optimize the thresholds to maximize the performance of the algorithm, without affecting the computation time.

Response 3: Supplementary explanations are given in the revised manuscript. You can find them in page 6 (191-201 lines).

(page 6)

2.5. Enhanced Pre-impact Fall Detection Algorithm

...

*In general, highly dynamic motions have large acceleration and angular velocity. However, vertical angles in ADLs are relatively different from those in fall motions. In present algorithm, two vertical angles (roll and pitch) were used. Roll and pitch angles were defined as the vertical angles of the body in the frontal and sagittal plane, respectively. Because the roll angle does not change significantly in ADLs, its threshold was set to |28°|. The threshold of pitch angle was set to 45°, since the upper body moved back and forth in some ADLs. As mentioned earlier, forward falls were not considered in this study, and thus, the absolute value of the pitch angle was not used in the algorithm. Figure 6 and 7 represent roll and pitch angles of less-dynamic, highly dynamic, and fall. In lying motions, vertical angle changed to about 90° as in fall motions, but acceleration SVM did not. Using the grid search method to maximize the specificity with 100% sensitivity, the threshold of acceleration SVM was set to 0.82 g.*

...

Point 4: In the abstract, the "lead time" is not defined. Just a short description, such as time lapse between the fall detection and the contact, would help a reader new in the field.

Response 4: I added supplementary explanations of the lead time in Abstract. (19 line in page 1)

(page 1)

Abstract

...

... lead time *(time lapse between the fall detection and the collision)* ...

...

Point 5: The Support Vector Machine could be defined as SVM, since it appears several times in the text (introduction). However, seeing that also Sum Vector Magnitude (SVM) is defined with the same acronym, it is reasonable to define only the second one which is more relevant for the paper. However, at the end of the introduction the acronym is defined twice.

Response 5: I revised them as your comments. (95, 104 lines in page 2, 3)

(page 2)

Introduction

...

Ahn et al. [23] developed a threshold-based fall detection algorithm based on acceleration *SVM*, angular velocity SVM, and triangle feature calculated from acceleration and angular velocity data from a sensor placed on the front waist. ...

(page 3)

...

From the captured inertial data, acceleration *SVM*, angular velocity SVM, and vertical angle were calculated using the complementary filter with a proportional integral (PI) controller.

...

Point 6: In section 2.2, it is not clear how the high-speed camera is used. Vicon is known for motion tracking, so it is not clear why using the Vicon if in the paper is not described any use of a motion tracking system. In Figure 2 for example, there are no markers on the subject, so it is reasonable to think that no motion tracking is used. If the Vicon camera is used not for motion tracking but for other purposes, it would be good to specify it explicitly.

Response 6: We only used a webcam and a Bonita high-speed camera (Vicon Motion System Ltd, UK) synchronized with the IMU sensor to determine the time when the body collided with the ground in fall motions. (123-125 lines in page 3)

(page 3)

2.2. Equipment

...

*Synchronized with the IMU sensor which was sampled at 100 Hz, a webcam and a Bonita high-speed camera (Vicon Motion System Ltd, UK) were used to determine the time when the body collided with the ground in fall motions.*

...

Round 2

Reviewer 2 Report

The authors addressed all the previous comments and concerns. Still, the selection of the thresholds could be done with a more elaborated method based on the analysis of the available data.